# The Burden of Associated Comorbidities in Patients with Obstructive Sleep Apnea—Regional Differences in Two Central–Eastern European Sleep Centers

**DOI:** 10.3390/jcm9113583

**Published:** 2020-11-06

**Authors:** Andras Bikov, Stefan Frent, Roxana Pleava, Laszlo Kunos, Saba Bokhari, Martina Meszaros, Stefan Mihaicuta

**Affiliations:** 1Department of Pulmonology, Semmelweis University, Budapest 1085, Hungary; andras.bikov@gmail.com (A.B.); laszlo.kunos@gmail.com (L.K.); martina.meszaros1015@gmail.com (M.M.); 2North West Lung Centre, Wythenshawe Hospital, Manchester University NHS Foundation Trust, Manchester M239LT, UK; saba.bokhari1@mft.nhs.uk; 3Division of Infection, Immunity & Respiratory Medicine, University of Manchester, Manchester M239LT, UK; 4Department of Pulmonology, University of Medicine and Pharmacy, Timisoara 300041, Romania; stefan.mihaicuta@umft.ro; 5Department of Cardiology, University of Medicine and Pharmacy, Timisoara 300041, Romania; roxana.pleava@gmail.com

**Keywords:** sleep apnea, comorbidities, cardiovascular disease, prevalence

## Abstract

Background: Obstructive sleep apnea (OSA) is usually associated with cardiovascular and cerebrovascular disease, metabolic syndrome and depression. Data on relevant OSA-associated comorbidities in Central–European populations are scarce. The aim of this study was to compare the prevalence of comorbidities in two OSA cohorts from Hungary and Romania. Methods: Data from 588 (282 from Hungary, 306 from Romania) untreated patients with OSA were retrospectively analyzed. The prevalence rates of hypertension, diabetes, dyslipidemia, allergic rhinitis, asthma, chronic obstructive pulmonary disease (COPD), osteoporosis, cerebrovascular and cardiovascular disease, arrhythmia and depression were compared between the two populations following adjustment for demographics, body mass index, smoking history, comorbidities and sleep parameters. Results: The prevalence rates of hypertension, arrhythmia, cerebrovascular and cardiovascular disease, diabetes and COPD in the whole study population were directly related to the severity of OSA. We found an inverse correlation between the prevalence of osteoporosis and OSA severity (all *p* < 0.05). Following adjustment, the prevalence of dyslipidemia was higher in the Hungarian cohort, whilst the prevalence of asthma, cardiovascular and cerebrovascular diseases was higher in the Romanian cohort (all *p* < 0.05). Conclusions: There was no difference in the prevalence rate of most comorbidities in patients with OSA from the two cohorts, except for dyslipidemia, asthma, cardiovascular and cerebrovascular disease.

## 1. Introduction

Obstructive sleep apnea (OSA) is a highly prevalent disease with a reported global prevalence of 1 billion people affected worldwide and estimated prevalence exceeding 50% in some countries [1]. It is characterized by a repetitive partial or total collapse of the upper airways during sleep, leading to chronic intermittent hypoxia and sleep fragmentation. Untreated OSA is associated with a significant morbidity burden including increased risk of motor vehicle accidents, neuropsychiatric dysfunction such as impaired cognition and depression, metabolic syndrome, cardiovascular and cerebrovascular disease and reduced life expectancy.

There may be a number of reasons why OSA is strongly associated with its recognized comorbidities. First, OSA and many of its associated comorbid conditions are common in the general adult population, and their co-occurrence is highly plausible. Second, older age, obesity and male gender are risk factors not only for OSA [2] but cardiometabolic diseases as well. Third, OSA-associated pathological processes such as hypoxemia, disturbed sleep, sympathetic surges, systemic inflammation, oxidative stress and hormonal changes can trigger or worsen cardiometabolic diseases [3] and depression [4]. Fourth, upper airway myopathy and autonomic neuropathy caused by metabolic disease may contribute to the increased severity of OSA as well [3].

There is a direct relationship between cardiometabolic and mood disorders including depression and OSA. However, the link between sleep apnea and respiratory diseases, such as allergic rhinitis, asthma and chronic obstructive pulmonary disease (COPD), is less obvious. Previously published data suggest that upper airway obstruction, upper and lower airway inflammation and reduced lung volumes may contribute to OSA, and co-existence of OSA and airway diseases may lead to a more marked overnight hypoxemia compared to each disease separately [5].

The strength of the association between OSA and its comorbidities depends on age, gender, obesity and smoking status [6,7]. Ethnicity is also likely to play a role [8]. More recently, we have reported that the relationship between hypertriglyceridemia and OSA may be linked to genetic factors [9], highlighting that there may be geographic variations in the prevalence of comorbidities in OSA. This is important to recognize, as it may inform the development of national policies on the screening for possible symptoms of OSA in patients with relevant comorbidities in targeted populations. The external phenotyping of the prevalence of associated comorbidities in OSA in different populations provides the opportunity to facilitate better individualized care.

The regional differences in the prevalence and clinical characteristics of a specific disease are an important question in many research studies. A population-based study performed in the region of Pomerania, northeastern Germany, investigated the risk factors and clinical diseases in a German cohort and reported a significant positive association of anthropometric measurements and self-reported cardiovascular diseases with obstructive sleep apnea [10]. Petrov I et al. conducted an observational study to explore the management of hyperlipidemia in high- and very high-risk patients from Central/Eastern Europe and Israel and found suboptimal levels of LDL-cholesterol in a significant proportion of treated patients [11]. Close collaboration across Europe is important, as learning from each other seems crucial to reducing health inequalities within Europe [12].

Data on the clinical characteristics of OSA, especially the burden of comorbidities in Central Europe, are scarce. According to a recent literature-based analysis, the estimated prevalence of OSA in Romania and Hungary in the adult population is 48.1% and 38.6%, respectively [1]. The aim of this retrospective analysis is to compare the profile of comorbidities in patients with OSA from these two neighboring countries in Central Europe.

## 2. Methods

### 2.1. Study Subjects and Design

We analyzed the data of patients from two OSA cohorts of consecutive patients enrolled between 2014 and 2019 in Budapest, Central Hungary, and Timisoara, Western Romania. Patients had either symptoms suggestive of sleep apnea (i.e., snoring, witnessed apnea, excessive daytime sleepiness, fatigue, etc.) or comorbidities frequently associated with OSA (i.e., obesity, cardiometabolic diseases, etc.). Data collection was performed in a systematic and unitary manner by trained sleep specialists, following the same protocol, in both countries. Demographic and anthropometric characteristics were collected from all patients as well as a detailed medical history, including Epworth Sleepiness Scale (ESS). Clinical data were collected using a standardized questionnaire which included a prespecified list of common OSA-associated comorbidities and prior to the confirmation of OSA. The list of comorbidities was based on patients’ reports, review of their current medications and available hospital notes. Cardiovascular and cerebrovascular diseases included current or previous stable and unstable angina, stroke, transient ischemic attack and significant atherosclerosis, and arrhythmias included supraventricular tachycardia and atrial fibrillation. This was followed by an inpatient sleep study. Patients not confirmed with OSA or who had been previously treated for OSA with upper airway surgery, a mandibular advancement device or continuous positive airway therapy were not included in the analysis. Patients with acute heart or renal failure and those with narcolepsy were excluded.

The studies were approved by the local Ethics Committees (Semmelweis University TUKEB 30/2014 and RKEB 172/2018 and 22/2014/24.07.2019 University of Medicine and Pharmacy Victor Babes Timisoara), and patients gave their informed consent before participating in the studies.

### 2.2. Sleep Studies

Inpatient polysomnography (all Romanian and 88 Hungarian patients) and cardiorespiratory polygraphy (194 Hungarian patients) were performed according to the American Academy of Sleep Medicine (AASM) recommendations [13]. Sleep stages and cardiorespiratory events were manually scored according to the AASM guidelines [14]. Apnea was defined as ≥90% drop in the nasal airflow lasting for ≥10 s. Hypopnea was defined as ≥30% drop in the nasal airflow lasting for ≥10 s accompanied with ≥3% drop in oxygen saturation (for polysomnography and polygraphy) or an arousal (for polysomnography). Patients attending inpatient cardiorespiratory polygraphy were monitored online with a video camera. Recording was started when they fell asleep. Cardiorespiratory and snore tracings helped to define sleep periods. Following the review of video recordings and the sleep study, periods of wakefulness were excluded from polygraphy analysis. We recorded total sleep time (TST), sleep period time (SPT) and minimal oxygen saturation (MinSatO2); sleep efficiency (Sleep%), percentage of rapid eye movement sleep phase (REM%), apnea–hypopnea index (AHI) and oxygen desaturation index (ODI) were calculated. AHI in REM (AHIREM) and non-REM (AHINREM) were also calculated separately. OSA was defined by an AHI ≥ 5/h.

### 2.3. Statistical Analyses

IBM SPSS Statistics (Version 23) was used for statistical analysis. The normality of the data was tested with the Kolmogorov–Smirnov test. Demographics and clinical characteristics of the two populations were compared with the *t*-test, Mann–Whitney and Chi-square tests. Unadjusted bivariate logistic regression analysis was used to study variables associated with comorbidities in the total cohort. To compare the burden of comorbidities between the two cohorts, unadjusted and multivariate logistic regression analyses were used. At the latter analysis, either traditional risk factors for OSA and comorbidities, such as age, gender, BMI, smoking status (ever/never) and AHI (model 1) were used as covariates, or those factors which were identified in the bivariate analysis (for instance, age, BMI, hypertension, diabetes, COPD, cerebro-/cardiovascular diseases, arrhythmia for dyslipidemia, model 2). Data were expressed as mean ± standard deviation or median/interquartile range. A *p* value < 0.05 was considered significant.

## 3. Results

### 3.1. Bivariate Analysis of Factors Associated with the Prevalence of Each Comorbidity

We identified a number of variables in the total study population which were significantly related to the prevalence of each comorbidity. The results are summarized in Table 1 and Table 2.

### 3.2. Comparative Analysis of Demographic and Clinical Characteristics of Patients from Hungary and Romania

Comparative data analysis of the two groups is summarized in Table 3.

In brief, Romanian patients were younger, had higher tobacco exposure, higher ESS, SPT, AHI, AHIREM, AHINREM, ODI and TST90% and had lower Sleep% and AI (all *p* < 0.05). There was also a trend toward a longer smoking history (*p* = 0.06) and a lower MinSatO2 (*p* = 0.08) in the Romanian cohort.

### 3.3. Comparison of the Prevalence of Comorbidities between Hungarian and Romanian Cohort

The raw prevalence of each comorbidity by country is illustrated in Figure 1.

Before adjustment, there was no difference in the prevalence of hypertension (*p* = 0.19), diabetes (*p* = 0.11), allergic rhinitis (*p* = 0.11), cardiac arrhythmia (*p* = 0.97) or depression (*p* = 0.40) between the two cohorts. Dyslipidemia (*p* < 0.01) and osteoporosis (*p* < 0.01) were more prevalent in Hungarian patients, while asthma (*p* < 0.01), COPD (*p* = 0.04) and cerebro-/cardiovascular disease (*p* < 0.01) were more common in Romanian patients.

Following adjustment for age, gender, BMI, smoking and AHI, the differences in the prevalence of dyslipidemia (*p* <0.01), asthma (*p* < 0.01), COPD (*p* = 0.01) and cerebro-/cardiovascular disease (*p* < 0.01) remained significant. In contrast, the prevalence of allergic rhinitis became significantly higher in the Hungarian cohort (*p* = 0.01). There were no significant differences between the countries after adjustment in the prevalence for other diseases (*p* > 0.05).

Following adjustment for covariates identified at the bivariate analysis, the prevalence rates of dyslipidemia (*p* < 0.01), asthma (*p* < 0.01), cerebrovascular and cardiovascular disease (*p* < 0.01) were significantly different. In contrast, the difference in the prevalence of allergic rhinitis (*p* = 0.35), COPD (*p* = 0.25) and osteoporosis (*p* = 0.28) became insignificant. Following adjustment according to model 2, there were no significant differences in the prevalence of other comorbidities between the two countries.

## 4. Discussion

This is the first study to explore the burden of comorbidities in Hungarian and Romanian patients with OSA. We found that dyslipidemia was more common in Hungary, whilst asthma, cerebrovascular and cardiovascular diseases were more prevalent in Romania.

We focused on the comorbidities that have previously been reported to be associated with an increased prevalence in OSA. However, only the prevalence of hypertension, arrhythmia, diabetes, COPD, cerebrovascular and cardiovascular disease were related to the severity of OSA in our study. In addition, we found an inverse relationship between the frequency of osteoporosis and AHI. This is in line with the report by Sforza et al. who hypothesized that intermittent hypoxia may stimulate bone remodeling [15]. The direct relationship between OSA severity and cardiometabolic diseases has previously been reported and is hypothesized to be due to chronic intermittent hypoxemia and sympathetic surges and in a lesser extent to sleep fragmentation [3,16]. In contrast, the relationship between OSA and COPD is less obvious [17]. Chronic intermittent hypoxia may inflict damages to the lung tissue, contributing to worsening of airflow limitation [5], and smaller lung volumes are associated with more severe OSA [18]. However, lung function parameters were not available in our study.

We did not find a relationship between OSA severity and other diseases, such as dyslipidemia, allergic rhinitis, asthma and depression. Dyslipidemia and depression were potentially underdiagnosed in our study, as we relied upon patients to self-report these conditions, and scrutiny was limited to the available medical records and drug charts. Allergic rhinitis in adulthood [19] and bronchial asthma [20] were not found to be risk factors for OSA; however, both may contribute to disease severity.

As part of the study, we distinguished between obstructive respiratory events occurring in REM and non-REM sleep. Recent attention has focused on the clinical significance of obstructive respiratory events occurring during REM sleep [21]. REM-related OSA was found to be an independent risk factor for cardiovascular disease [22] and diabetes [23], although no such association was found for dyslipidemia [24]. REM sleep is characterized by increased sympathetic activity and decreased muscle tone compared to non-REM sleep, and in addition, there is an altered production of some hormones such as the growth hormone or thyrotropin between the two phases of sleep [21]. However, in our study, we did not find a statistically significant difference between AHIREM and AHINREM sleep phases in the prevalence of comorbidities.

Our study identified statistically significant differences in some clinical variables between Hungarian and Romanian patients diagnosed with OSA. Such heterogeneity has previously been observed from the European Sleep Apnoea Database (ESADA) [25]. In our study, Romanian patients with OSA had a higher disease severity than their Hungarian counterparts and reported worse sleep efficiency and increased daytime sleepiness which would be in keeping with more severe disease. However, in our cohort, 67% of Hungarian patients underwent a polygraphy instead of a full polysomnography. As the definition of hypopnea on polysomnography includes arousals even in the absence of desaturations, polysomnography is likely to yield a higher AHI than polygraphy, and this could partly be responsible for the observed differences in disease severity between the two cohorts.

We found significant differences in the prevalence of dyslipidemia, asthma and cardiovascular/cerebrovascular disease between the two countries following adjustment on covariates. Dyslipidemia is related to OSA via multiple direct mechanisms (chronic intermittent hypoxia, sympathetic bursts, hormonal changes) and common a etiology (i.e., obesity) [16]. However, interestingly, we did not find a significant association between OSA severity and the prevalence of dyslipidemia. Previous studies reported only a weak relationship between serum lipid levels and markers of OSA severity [26,27]. We have previously reported on this association and suggested that this relationship may be linked to genetic factors [9]. However, this is unlikely to be the reason for the international differences noted in this study, as the genetic background of the population residing in Western Transylvania is very similar to the Hungarian one [28]. Dyslipidemia was related to age and BMI in our study, but the difference remained statistically significant even after adjustment for these confounding factors. A possible explanation for this likely resides in the differences in diet and physical activity in the studied populations, although we do not have any data to support/refute this. It is possible that Hungarian patients may have been more rigidly screened for dyslipidemia compared to those from Romania. Asthma was more prevalent amongst Romanian patients with OSA. This is not surprising given that the overall prevalence and severity of asthma in Central and Eastern Europe is highest in Romania [29].

The higher prevalence of cardiovascular and cerebrovascular disease in Romanian patients was somewhat surprising. These diseases were directly related to age, BMI, cigarette pack years, hypertension, diabetes, dyslipidemia and OSA severity in our study. Although there were differences in some of these variables between the two countries, the prevalence of cerebrovascular and cardiovascular disease was significantly higher in Romanian patients even following adjustment for these factors. Although the data on cardiovascular and cerebrovascular morbidity in central Europe are scarce, the age-standardized cardiovascular mortality in the two countries is similar [30,31]. In both cohorts, patients were referred for a diagnostic sleep test based on symptoms and comorbidities. The observed discrepancy could possibly be explained by different levels of OSA awareness among the referring cardiologists and internists between the two countries. In line with this, a significant geographical and interdisciplinary variance in OSA awareness was also noticed in other reports [32]. Of note, the prevalence of cerebrovascular and cardiovascular disease in both populations was higher compared to the general population (around 10%) [33]. However, both values were still lower compared to the asymptomatic coronary plaque burden (80%) as reported in a previous study [34]. Apart from risk factors accounted for in our study, additional factors including alcohol consumption, diet, exercise and air pollution could contribute to the observed differences. Unfortunately, data on these additional factors were not available to us.

We recognize this study has a number of limitations. Comorbidities were defined according to the reports of the participants, current medications and hospital charts. Some of these reports may have been inaccurate and some diseases may have been underdiagnosed. The comorbidities relied on self-reporting by the patients who may not have considered/wanted to disclose certain conditions. In addition, polysomnography was performed in only one third of the Hungarian patients compared to 100% in the Romanian patients. Both polysomnography and cardiorespiratory polygraphy are validated tools for the detection of sleep apnea [35] and are used in research purposes for data collection in other large cohorts of sleep apnea patients, such as ESADA [25]. As previously described, this discrepancy is likely to have led to an underreporting of OSA severity in the Hungarian cohort. Although the primary aim of the study was to compare the burden of comorbidities, Model 1 and some of the analyses in Model 2 were adjusted for AHI; therefore, those conclusions need to be interpreted with caution. Notably, AHI already has a poor inter-night reproducibility due to various factors (sleep stage, position, diet, etc.) [36]. This was a retrospective analysis of data that had already been collected so it is prone to bias. Environmental factors, such as diet, exercise, alcohol consumption and air pollution were not available to us during our analysis which may have helped us to substantiate/refute these factors as causing the observed differences between the two cohorts. Lastly, we investigated only patients with OSA. Data on primary snorers as controls could have helped in understanding the burden of comorbidities in OSA in more detail, although it is recognized that the prevalence of certain comorbidities in OSA is not necessarily indicative of prognosis from OSA.

## 5. Conclusions

In summary, this was the first retrospective study describing the profile of comorbidities in patients with OSA in Hungary and Romania. There was no difference in the prevalence for most comorbidities between the two countries. In contrast, dyslipidemia was more prevalent in Hungary and asthma, cerebrovascular and cardiovascular diseases were more common in Romania. We believe that our data are valuable for policy makers when designing patient pathways and identifying possible areas (as guided by the local prevalence of associated comorbidities) on which to focus the screening for sleep-related symptoms in the diagnosis of OSA.

## Figures and Tables

**Figure 1 jcm-09-03583-f001:**
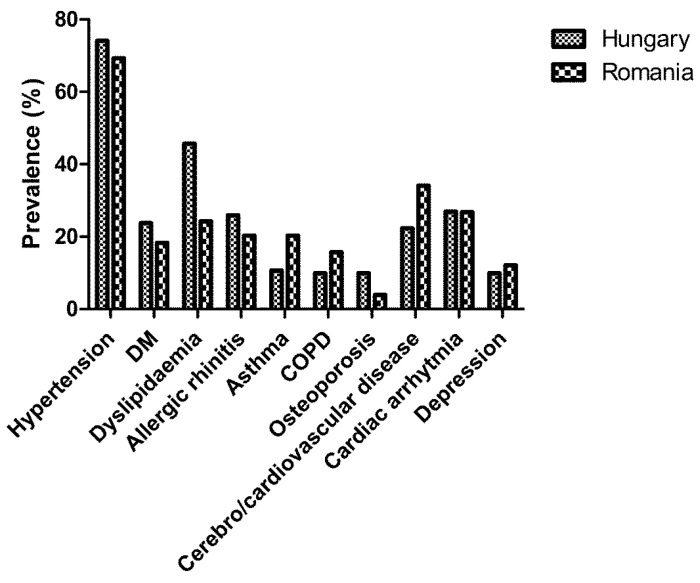
Comparative prevalence of comorbidities in the Hungarian and Romanian cohorts.

**Table 1 jcm-09-03583-t001:** Regression coefficients (β) of the bivariate logistic regression between comorbidities and clinical characteristics (N = 588).

	Hypertension	Diabetes	Dyslipidemia	COPD	Allergic Rhinitis	Asthma	Cerebro-/Cardiovascular Disease	Arrhythmia	Osteoporosis	Depression
Age	**0.078**	**0.051**	**0.036**	0.018	**−0.032**	**−0.019**	**0.071**	**0.055**	**0.110**	0.018
Gender (male)	**0.480**	**0.591**	0.318	**−0.870**	0.010	**0.542**	0.020	**0.474**	**2.203**	**0.738**
BMI	**0.118**	**0.076**	**0.039**	**0.059**	−0.015	**−0.001**	**0.034**	**0.042**	**−0.097**	0.010
History of smoking	0.129	−0.096	−0.170	**2.536**	−0.113	**−0.752**	0.304	−0.091	**−1.171**	−0.231
PY	**0.016**	**0.012**	0.003	**0.089**	**−0.019**	**−0.026**	**0.022**	0.010	**−0.041**	0.001
Hypertension	-	**1.637**	**1.210**	**1.677**	**−0.824**	**−1.022**	**1.925**	**1.417**	0.333	0.218
Diabetes	**1.637**	-	**1.459**	0.506	**−0.526**	0.058	**1.040**	**0.539**	0.390	−0.291
Dyslipidemia	**1.210**	**1.459**	-	**0.558**	−0.380	−0.343	**0.952**	**0.611**	0.582	0.192
COPD	**1.677**	0.506	**0.558**	-	**−2.134**	**−1.067**	**1.212**	**0.993**	−0.308	−0.634
Allergic rhinitis	**−0.824**	**−0.526**	−0.380	**−2.134**	-	**1.005**	**−0.529**	**−0.487**	0.393	−0.196
Asthma	**−1.022**	0.058	−0.343	**−1.067**	**1.005**	-	−0.419	−0.403	0.486	−0.160
Cerebro-/cardiovascular disease	**1.925**	**1.040**	**0.952**	**1.212**	**−0.529**	−0.419	-	**1.276**	0.206	0.363
Arrhythmia	**0.270**	**0.539**	**0.611**	**0.993**	**−0.487**	−0.403	**1.276**	-	**0.758**	0.132
Osteoporosis	0.333	0.390	0.582	−0.308	0.393	0.486	0.206	0.758	-	**0.936**
Depression	0.218	−0.291	0.192	−0.634	−0.196	−0.160	0.363	0.132	**0.936**	-

BMI—body mass index; COPD—chronic obstructive pulmonary disease; significant relationships are shown in bold.

**Table 2 jcm-09-03583-t002:** Regression coefficients (β) of the bivariate logistic regression between comorbidities and sleep parameters (N = 588).

	Hypertension	Diabetes	Dyslipidemia	COPD	Allergic Rhinitis	Asthma	Cerebro-/Cardiovascular Disease	Arrhythmia	Osteoporosis	Depression
AHI	**0.010**	**0.010**	−0.004	**0.015**	−0.001	0.003	**0.009**	**0.011**	**−0.021**	−0.007
AHI_REM_	0.000	0.000	0.000	0.000	−0.005	0.000	0.000	0.000	**−0.018**	0.000
AHI_NREM_	**0.011**	**0.015**	−0.002	**0.015**	−0.004	0.004	**0.012**	**0.009**	**−0.019**	−0.004
ODI	**0.012**	**0.010**	−0.003	**0.015**	−0.002	0.003	**0.010**	**0.012**	**−0.022**	−0.005
TST90%	**0.004**	**0.005**	0.000	**0.007**	−0.001	0.001	**0.005**	**0.005**	0.000	0.000
MinSatO_2_	**−0.047**	**−0.040**	−0.011	**−0.044**	**0.029**	−0.003	**−0.026**	**−0.033**	0.004	0.016
AI	0.003	0.005	0.007	0.006	0.003	−0.011	0.002	0.008	0.018	**−0.048**
ESS	0.018	0.041	0.000	**0.088**	−0.001	0.029	**0.067**	0.019	−0.053	0.028

AHI—apnea–hypopnea index; AHI_NREM_—apnea–hypopnea index in non-REM (NREM) sleep; AHI_REM_—apnea–hypopnea index in REM sleep; AI—arousal index; COPD—chronic obstructive pulmonary disease; ESS—Epworth Sleepiness Scale; MinSatO2—minimal oxygen saturation; ODI—oxygen desaturation index; TST90%—% of sleep spent with oxygen saturation under 90%. Significant relationships are shown in bold.

**Table 3 jcm-09-03583-t003:** Comparison of demographic, clinical characteristics and sleep parameters of Hungarian and Romanian patients with obstructive sleep apnea (OSA).

	Hungary (n = 282)	Romania (n = 306)	*p*
Age	59 /48–67/	52 /44–58/	<0.01
Gender (male%)	66.3	71.6	0.17
BMI	32.1 /17.3–37.3/	32.2 /29.0–35.9/	0.72
Smoking status (ever/never)	120/162	154/152	0.06
Smoking status (current/ex/never)	79/41/162	91/63/152	0.09
Cigarette pack years	0 /0–12/	0 /0–19/	0.02
ESS	6 /3–9/	10 /6–13/	<0.01
TST	415 /380–445/ *	421 /360–466/	0.26
SPT	449 /413–483/ *	479 /443–512/	<0.01
Sleep%	95 /89–97/ *	90 /79–96/	<0.01
REM%	16.4 /12.0–21.2/ *	14.9 /11.5–20.1/	0.32
AHI	22.7 /12.2–42.4/	39.3 /23.6–63.3/	<0.01
AHI_REM_	24.6 /11.8–46.2/ *	41.1 /23.9–62.3/	<0.01
AHI_NREM_	16.7 /9.6–30.9/ *	38.6 /22.5–62.2/	<0.01
ODI	19.35 /9.4–35.8/	38.0 /22.0–61.3/	<0.01
TST90%	4.2 /0.7–13.8/	13.0 /3.0–52.5/	<0.01
MinSatO_2_	83 /76–86/	81 /72–87/	0.08
AI	43.4 /29.6–59.4/ *	18.3 /11.2–26.5/	<0.01

AHI—apnea–hypopnea index; AHI_NREM_—apnea–hypopnea index in NREM sleep; AHI_REM_—apnea–hypopnea index in REM sleep; AI—arousal index; BMI—body mass index; COPD—chronic obstructive pulmonary disease; ESS—Epworth Sleepiness Scale; MinSatO_2_—minimal oxygen saturation; ODI—oxygen desaturation index; REM%—% of REM sleep; Sleep%—sleep efficiency; SPT—sleep period time; TST—total sleep time; TST90%—% of sleep spent with oxygen saturation under 90%. * Polysomnography has been performed for 88 Hungarian patients.

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
