# Peer review of "The Burden of Associated Comorbidities in Patients with Obstructive Sleep Apnea—Regional Differences in Two Central–Eastern European Sleep Centers"

_jcm, 2020, doi:10.3390/jcm9113583_

Round 1
Reviewer 1 Report
It is very difficult to re-review this paper when the authors have not provided a letter responding to the significant concerns from both reviewers.
Were the data retrospectively collected? Line 97 reads as though it was prospectively collected. Please clarify. More information is needed on the sample - how were the samples selected? was it all patients who attended for a sleep study between certain dates, for example?
The results of the multivariate analyses are difficult to read. Authors should consider presenting them in a table, and providing the full results of the multivariate analyses in an online supplement
Also consider including the "standardised questionnaire" and protocol used to collect the comorbidity data in an online supplement
Table 1 and 2 - assume this is the total sample (ie n=588)?
How was sleep staged in those patients who only underwent polygraphy?
I have difficulty believing the conclusions given they are based the relationship between OSA severity and comorbidities, and yet OSA severity was not measured uniformly across both samples.
Author Response
Dear Reviewer,
Please find attached our letter of responses to the raised issues.
Kind regards,
dr. Stefan Frent

Reviewer 2 Report
The authors made a retrospective study concerning OSA comorbidities. Minor comments:
- The manuscript should be reviewed by a native English speaker.
- Table 3 highlights a major concern: the authors did not analyze the same kind of patients in Hungary and Roumania (AHI, ESS, ODI, AI, ...). This is a huge limitation.
- In 2020, the authors can't publish a manuscript dealing with OSA, without a polysomnography for each patient. In that what, the two-third of Hungarian patients should be removed and statistics remake.
- Patients should be classified as mild, moderate, or severe OSA as these three categories have a different impact on the patient's health.
Author Response
Dear Reviewer,
Please find attached our letter of responses to the raised issued.
Kind regards,
dr. Stefan Frent

Round 2
Reviewer 1 Report
Thank you. My concerns have been addressed and I am pleased to recommend this paper for publication. I am unable to open the .rar supplementary files, but will leave this to the editors to determine their suitability for inclusion.
Reviewer 2 Report
Unfortunately, the authors failed to improve their manuscript.
The same limitations are still present and made for me the manuscript unacceptable in 2020. Maybe the authors can present it as a poster in an International Meeting.
This manuscript is a resubmission of an earlier submission. The following is a list of the peer review reports and author responses from that submission.
Round 1
Reviewer 1 Report
Unfortunately I am unable to recommend this paper for publication due to some major flaws with the rationale and methodology.
I don't understand why the authors chose to investigate the difference between the prevalence of comorbidities in OSA cohorts in Romania and Hungary. There is a very weak justification in the introduction that I did not find at all convincing (genetic differences) and even the authors report the flaws of this rationale in the discussion.
Secondly, the method for detecting OSA between the 2 cohorts was different (polygraphy vs polysomnography) which are well know to detect different amounts of OSA. This undermines the entire study.
There is not enough information provided about how the participants were sampled (was it the same in both countries?), how the comorbidities were defined and identified, who collected the data (were they blinded to OSA results? How did you ensure it was done in the same way in both cohorts?) I am not confident at all that the methods were standardised enough to be able to make sensible comparisons between the cohorts.
I found the regression modelling confusing, perhaps because I found the research question confusing.
Reviewer 2 Report
Dear authors,
First of all congratulations for your work.
Despite of OSA is a topic that is very analysed in the literature, it is interesting to have data and publications about it in the countries of East-Europe, so I agree with the idea to be pioneer and to announce results to the scientific public. I also like the sample is too large (588 patients).
However, there are lots of limitations (inaccurate collect data, lack of polysomnography...) which it is very difficult to conclude with strictness.
So, I am sorry but the power of this study is not high enough.
I appreciate the time and effort it takes to submit your work and hope that this decision does not deter you from submitting to JCM in the future.